# On Supervised Classification of Feature Vectors with Independent and Non-Identically Distributed Elements

**DOI:** 10.3390/e23081045

**Published:** 2021-08-13

**Authors:** Farzad Shahrivari, Nikola Zlatanov

**Affiliations:** Electrical and Computer Systems Engineering, Monash University, Alliance Ln, Clayton, VIC 3168, Australia; Farzad.shahrivari@monash.edu

**Keywords:** supervised classification, independent and non-identically distributed features, analytical error probability

## Abstract

In this paper, we investigate the problem of classifying feature vectors with mutually independent but non-identically distributed elements that take values from a finite alphabet set. First, we show the importance of this problem. Next, we propose a classifier and derive an analytical upper bound on its error probability. We show that the error probability moves to zero as the length of the feature vectors grows, even when there is only one training feature vector per label available. Thereby, we show that for this important problem at least one asymptotically optimal classifier exists. Finally, we provide numerical examples where we show that the performance of the proposed classifier outperforms conventional classification algorithms when the number of training data is small and the length of the feature vectors is sufficiently high.

## 1. Introduction

### 1.1. Background

Supervised classification is a machine learning technique that maps an input feature vector to an output label based on a set of correctly labeled training data. There is no single learning algorithm that works best on all supervised learning problems, as shown by the no free lunch theorem in [1]. As a result, there are many algorithms proposed in the literature whose performance depends on the underlying problem and the amount of training data available. The most widely used algorithms in the literature are decision trees [2,3], Support Vector Machines (SVM) [4,5], Rule-Based Systems [6], naive Bayes classifiers [7], k-nearest neighbors (KNN) [8], logistic regressions, and neural networks [9,10].

### 1.2. Motivation

In the following, we discuss the motivation for this work.

#### 1.2.1. Lack of Tight Upper Bounds on the Performance of Classifiers

In general, there are no tight upper bounds on the performance of the classifiers used in practice. Many of the previous works only provide experimental performance results. However, this approach has drawbacks. For example, one has to rely on the trial-and-error approach in order to develop a good classifier for a given problem, which impacts the reliability. Next, the algorithms whose performance has been verified only experimentally may work for a given problem, but may fail to work when applied to a similar problem. Finally, experimental results do not provide intuition into the underlying problem, whereas the analytical results provide the understanding of the underlying problem and the corresponding solutions.

Motivated by this, in the paper, we aim to investigate classifiers with analytical upper bounds on their performance.

#### 1.2.2. Independent and Non-Identically Distributed Features

In general, we can categorize the statistical properties of the feature vectors, which are the input to the classifier, into three types. To this end, let $Y^{n}\left(X\right)=\left[Y_{1}\left(X\right),Y_{2}\left(X\right),\ldots ,Y_{n}\left(X\right)\right]$ denote the input feature vector to the supervised classifier, where *n* is the length of the feature vector and *X* is the label to which the feature vector $Y^{n}\left(X\right)$ belongs. Then, we can distinguish the following three types of feature vectors depending on the statistics of the elements in the feature vector $Y^{n}\left(X\right)$.

The first type of feature vector is when the elements of $Y^{n}\left(X\right)$ are independent and identically distributed (i.i.d.). This is the simplest features model, but also the least applicable in practice. This model is identical to hypothesis testing, which has been well investigated in the literature [11,12,13]. As a result, tight upper bounds on the performance of supervised learning algorithms for this type of feature vector are available in the hypothesis testing literature. For instance, the authors in [11] showed that the posterior entropy and the maximum a posterior error probability decay to zero with the length of the feature vector at the identical exponential rate, where the maximum achievable exponent is the minimum Chernoff information. In [12], the authors determine the requirements for the length of the vector $Y^{n}\left(X\right)$ and the number of labels *m* in order to achieve vanishing exponential error probability in testing *m* hypothesis that minimizes the rejection zone. In [13], the authors provide an upper bound and a lower-bound on the error probability of Bayesian *m*-ary hypothesis testing in terms of conditional entropy.

The second type of feature vectors is when the elements of $Y^{n}\left(X\right)$ are mutually dependent and non-identically distributed (d.non-i.d.). This type of features model is the most general model and the most applicable in practice. However, it is also the most difficult to tackle analytically. As a result, supervised learning algorithms proposed for this features model lack analytical tight upper bounds on their performance [14,15,16,17,18,19,20,21,22,23]. This is because there are not any frameworks that produce closed-form results when deriving statistics of vectors with d.non-i.d. elements when the underlying distributions are unknown. Then how can we investigate analytically classifiers for practical scenarios when the feature vectors have d.non-i.d. elements? A possible approach leads us to the third type of feature vectors, explained in the following.

The third type of feature vectors is when the elements of $Y^{n}\left(X\right)$ are mutually independent but non-identically distributed (i.non-i.d.). This features model is much simpler than the d.non-i.d. features model and, more importantly, it is analytically tractable, as we show in this paper. Furthermore, this features model is applicable in practice. Specifically, there exists a class of algorithms, known as Independent Component Analysis (ICA), that transform vectors with d.non-i.d. elements into vectors with i.non-i.d. elements with a zero or a negligible loss of information [24,25,26,27,28]. The origins of ICA can be traced back to Barlow [29], who argued that a good representation of binary data can be achieved by an invertible transformation that transform vectors with d.non-i.d. elements into vectors with i.non-i.d. elements. Finding such a transformation with no prior information about the distribution of the data has been considered an open problem until recently [28]. Specifically, the authors in [28] show that this hard problem can be accurately solved with a branch and bound search tree algorithm, or tightly approximated with a series of linear problems. Thereby, the authors in [28] provide the first efficient set of solutions to Barlow’s problem. So far, the complexity of the fastest such algorithm is $O\left(n\times 2^{n}\right)$ [28]. Nevertheless, since there exist such invertible transformations (i.e., no loss of information) which can transform vectors with d.non-i.d. elements into vectors with i.non-i.d. elements, we can tackle the features model comprised of d.non-i.d. elements by first transforming it (without loss of information) into the features model comprised of i.non-i.d. elements and then tackling the i.non-i.d. features model.

Motivated by this, in this paper, we investigate supervised classification of feature vectors with i.non-i.d. elements.

#### 1.2.3. Small Training Set

The main factor that impacts the accuracy of supervised classification is the amount of training data. In fact, most supervised algorithms are able to learn only if there is a very large set of training data available [30]. The main reason for this is the curse of dimensionality [31,32], which states that “the higher the dimensionality of the feature vectors, the more training data are needed for the supervised classifier” [33]. For example, supervised classification methods such as random forest [34,35] and KNN [36] suffer from the curse of dimensionality. However, having large training data sets is not always possible in practice. As a result, designing a supervised classification algorithm that exhibits good performance even when the training data set is extremely small is important.

Motivated by this, in this paper, we investigate supervised classifiers for the case when *t* training feature vectors per label are available, where t=1,2,...

### 1.3. Contributions

In this paper, we propose an algorithm for supervised classification of feature vectors with i.non-i.d. elements when the number of training feature vectors per label is *t*, where t=1,2,... Next, we derive an upper bound on the error probability of the proposed classifier for uniformly distributed labels and prove that the error probability exponentially decays to zero when the length of the feature vector, *n*, grows, even when only one training vector per label is available, i.e., when t=1. Hence, the proposed classification algorithm provides an asymptotically optimal performance even when the number of training vectors per label is extremely small. We compare the performance of the proposed classifier with the naive Bayes classifier and to the KNN algorithm. Our numerical results show that the proposed classifier significantly outperforms the naive Bayes classifier and the KNN algorithm when the number of training feature vectors per label is small and the length of the feature vectors *n* is sufficiently high.

The proposed algorithm is a form of the nearest neighbor classification algorithm, where the nearest neighbor is searched in the domain of empirical distributions. As a result, we refer to the algorithm as *the nearest empirical distribution*. The nearest empirical distribution algorithm is not new and, to the best of our knowledge, it was first proposed in [37] for the case when the elements of $Y^{n}\left(X\right)$ are i.i.d., i.e., for the equivalent problem of hypothesis testing. However, in this paper, we propose the nearest empirical distribution algorithm for the case when the elements of $Y^{n}\left(X\right)$ are i.non-i.d., which is much more complex than the problem of hypothesis testing where the elements of $Y^{n}\left(X\right)$ are i.i.d.

To the best of our knowledge, this is the first paper that investigates the important problem of classifying feature vectors with i.non-i.d. elements and provides an upper bound on its error probability. The novelty of this paper is not with the classifier itself, but rather in showing the importance of the problem of classifying feature vectors with i.non-i.d elements and in showing analytically that at least one classifier with an asymptotically optimal error probability exists when at least one training feature vectors per label is available.

The remainder of this paper is structured as follows. In Section 2, we formulate the considered classification problem. In Section 3, we provide our classifier and derive an upper bound on its error probability. In Section 4, we provide numerical examples of the performance on the proposed classifier. Finally, Section 5 concludes the paper.

## 2. Problem Formulation

The machine learning model is comprised of a label *X*, a feature vector $Y^{n}\left(X\right)=[Y_{1}\left(X\right),Y_{2}\left(X\right),$$\ldots ,Y_{n}\left(X\right)]$ of length *n* mapped to the label *X*, and a learned label $\hat{X}$, as shown in Figure 1. In this paper, we adopt the information-theoretic style of notations and thereby random variables are denoted by capital letters and their realizations are denoted with small letters. The feature vector $Y^{n}\left(X\right)$ is the input to the machine learning algorithm whose aim is to detect the label *X* from the observed feature vector $Y^{n}\left(X\right)$. The performance of the machine learning algorithm is measured by the error probability $P_{e}=\mathrm{Pr}\left\{X\neq \hat{X}\right\}$.

We adopt the modeling in [38,39,40] and represent the dependency between the label *X* and the feature vector $Y^{n}\left(X\right)$ via a joint probability distribution $p_{X,Y^{n}}\left(x,y^{n}\right)$. Now, in order to gain a better understanding of the problem, we include the joint probability distribution $p_{X,Y^{n}}\left(x,y^{n}\right)$ into the model in Figure 1. To this end, since $p_{X,Y^{n}}\left(x,y^{n}\right)=p_{Y^{n}|X}\left(y^{n}|x\right)p_{X}\left(x\right)$ holds, instead of $p_{X,Y^{n}}\left(x,y^{n}\right)$, we can include the conditional probability distribution $p_{Y^{n}|X}\left(y^{n}|x\right)$ and the probability distribution $p_{X}\left(x\right)$ into the model in Figure 1, and thereby obtain the model in Figure 2.

Now, the classification learning model in Figure 2 is a system comprised of a label generating source *X* according to the distribution $p_{{}_{X}}\left(x\right)$, a feature vector generator modelled by the conditional probability distribution $p_{{}_{Y^{n}|X}}\left(y^{n}|x\right)$, a feature vector $Y^{n}$, a classifier that aims to detect *X* from the observed feature vector $Y^{n}$, and the detected label $\hat{X}$. Note that the system model in Figure 2 can be seen equivalently as a communication system comprised of a source *X*, a channel with input *X* and output $Y^{n}$, and a decoder (i.e., detector) that aims to detect *X* from $Y^{n}$. The notation used in this paper, letter *X* for labels and letter *Y* for features, is based on the notation used in information theory for modelling communication systems. In the classification model shown in Figure 2, we assume that the label *X* can take values from the set X, according to $p_{{}_{X}}\left(x\right)=1/\left|X\right|$, where |·| denotes the cardinality of a set. Next, we assume that the *i*-th element of the feature vector $Y^{n}$, $Y_{i}$, for i=1,2,⋯,n, takes values from the set $Y=\left\{y_{1},y_{2},\ldots ,y_{\left|Y\right|}\right\}$, according to the conditional probability distribution $p_{{}_{Y_{i}|X}}\left(y_{i}|x\right)$.

Moreover, we assume that the elements of the feature vector $Y^{n}$ are i.non-i.d. As a result, the feature vector $Y^{n}$ takes values from the set $Y^{n}$ according to the conditional probability distribution $p_{{}_{Y^{n}|X}}\left(y^{n}|x\right)$ given by
(1)$$p_{{}_{Y^{n}|X}}\left(y^{n}|x\right)=p_{{}_{Y_{1},Y_{2},\ldots ,Y_{n}|X}}\left(y_{1},y_{2},\ldots ,y_{n}|x\right)\overset{\left(a\right)}{=}\prod_{i=1}^{n}p_{{}_{Y_{i}|X}}\left(y_{i}|x\right)\overset{\left(b\right)}{=}\prod_{i=1}^{n}p_{{}_{i}}\left(y_{i}|x\right),$$
where (a) comes from the fact that elements in the feature vector $Y^{n}$ are mutually independent and (b) is for the sake of notational simplicity, where $p_{{}_{i}}$ is used instead of $p_{{}_{Y_{i}|X}}$. As a result of (Equation 1), the considered classification model in Figure 2 can be represented equivalently as in Figure 3.

Next, we assume that $p_{{}_{i}}\left(y_{i}|x\right)$, ∀i, and thereby $p_{{}_{Y^{n}|X}}\left(y^{n}|x\right)$ are unknown to the classifier. Instead, the classifier knows X, Y, and for each $x_{i}\in X$, where i=1,2,…,|X|, it has access to a finite set of *t* correctly labelled input–output pairs $\left(x_{i},{\hat{y}}_{i_{1}}^{n}\right),\left(x_{i},{\hat{y}}_{i_{2}}^{n}\right),\ldots ,$
$\left(x_{i},{\hat{y}}_{i_{t}}^{n}\right)$, denoted by $T_{i}$, referred to as the training set for label $x_{i}$.

Finally, we assume that the following holds
(2)$$\sum_{l=1}^{n}p_{l}\left(y|x_{i}\right)\neq \sum_{l=1}^{n}p_{l}\left(y|x_{j}\right),\;\mathrm{for}\;y\in Y\;\mathrm{and}\;i\neq j.$$

The condition in (Equation 2) means that the distribution of the feature vectors $Y^{n}\left(X\right)$ for label X=i is not a perturbation of distribution of the feature vectors $Y^{n}\left(X\right)$ for label X=j. As a result, the proposed classifier only applies to the subset of data vectors with i.non-i.d. elements that satisfy (Equation 2).

For the classification system model defined above and illustrated in Figure 3, we wish to propose a classifier that exhibits an asymptotically optimal error probability $P_{e}=\mathrm{Pr}\left\{X\neq \hat{X}\right\}$ with respect to the length of $Y^{n}$, *n*, for any t≥1, i.e., for any t≥1, $P_{e}\to 0$ as n→∞. Moreover, we wish to obtain an analytical upper bound on the error probability of the proposed classifier for a given *t* and *n*.

## 3. The Proposed Classifier and Its Performance

In this section, we propose our classifier, derive an analytical upper bound on its error probability, and prove that the classifier exhibits an asymptotically optimal performance when the length of the feature vector $Y^{n}$, *n*, satisfies n→∞. This is conducted in the following.

For a given vector $v^{n}=\left(v_{1},v_{2},\ldots ,v_{n}\right)$, let the Minkowski distance *r* be defined as
(3)$$\begin{array}{c} {∥v∥}_{r}={\left(\sum_{i=1}^{n}v_{i}^{r}\right)}^{\left(1/r\right)}. \end{array}$$

Moreover, for a given feature vector $y^{k}=\left(y_{1},y_{2},,\ldots ,y_{k}\right)$, let $I[y^{k}=y]$ be a function defined as
(4)$$\begin{array}{c} I\left[y^{k}=y\right]=\sum_{i=1}^{k}Z\left[y_{i}=y\right], \end{array}$$
where $Z[y_{i}=y]$ is an indicator function assuming the value 1 if $y_{i}=y$ and 0 otherwise. Hence, $I[y^{k}=y]$ counts the number of elements in $Y^{k}$ that have the value *y*.

### 3.1. The Proposed Classifier

Let ${\hat{y}}_{i}^{nt}$ be a vector obtained by concatenating all training feature vectors for the input label $x_{i}$ as
(5)$$\begin{array}{c} {\hat{y}}_{i}^{nt}=\left({\hat{y}}_{i_{1}}^{n},{\hat{y}}_{i_{2}}^{n},\ldots ,{\hat{y}}_{i_{t}}^{n}\right). \end{array}$$

Let $P_{{\hat{y}}_{i}^{nt}}$ be the empirical probability distribution of the concatenated training feature vector for label $x_{i}$, ${\hat{y}}_{i}^{nt}$, given by
(6)$$\begin{array}{c} P_{{\hat{y}}_{i}^{nt}}=\left[\frac{I\left[{\hat{y}}_{i}^{nt}=y_{1}\right]}{nt},\frac{I\left[{\hat{y}}_{i}^{nt}=y_{2}\right]}{nt},\ldots ,\frac{I\left[{\hat{y}}_{i}^{nt}=y_{\left|Y\right|}\right]}{nt}\right]. \end{array}$$

Let $y^{n}$ be the observed feature vector at the classifier whose label it wants to detect and let $P_{y^{n}}$ denote the empirical probability distribution of $y^{n}$, given by
(7)$$\begin{array}{c} P_{y^{n}}=\left[\frac{I\left[y^{n}=y_{1}\right]}{n},\frac{I\left[y^{n}=y_{2}\right]}{n},\ldots ,\frac{I\left[y^{n}=y_{\left|Y\right|}\right]}{n}\right]. \end{array}$$

Using the above notations, we propose the following classifier.

**Proposition** **1.**
*For the considered system model, we propose a classifier with the following classification rule*
(8)$$\begin{array}{c} \hat{x}=x_{i},\;\mathrm{where}\;i=\arg\min_{i}{∥P_{y^{n}}-P_{{\hat{y}}_{i}^{nt}}∥}_{r}, \end{array}$$
*where r≥1 and ties are resolved by assigning the label among the ties uniformly at random. (For example, if ${∥P_{y^{n}}-P_{{\hat{y}}_{i}^{nt}}∥}_{r}={∥P_{y^{n}}-P_{{\hat{y}}_{j}^{nt}}∥}_{r}$ holds for, i≠j, we set $\hat{x}=x_{i}$ or $\hat{x}=x_{j}$ uniformly at random).*


As seen from (Equation 8), the proposed classifier assigns the label $x_{i}$ if the empirical probability distribution of the concatenated training feature vector mapped to label $x_{i}$, $P_{{\hat{y}}_{i}^{nt}}$ is the closest, in terms of Minkowski distance *r*, to the empirical probability distribution of the observed feature vector $P_{y^{n}}$. In that sense, the proposed classifier can be considered as the nearest empirical distribution classifier.

### 3.2. Upper Bound on the Error Probability

The following theorem establishes an upper bound on the error probability of the proposed classifier.

**Theorem** **1.**
*Let ${\bar{P}}_{j}$, for j=1,2,…,|X|, be a vector defined as*
(9)$$\begin{array}{c} {\bar{P}}_{j}=\left[\bar{p}\left(y_{1}|x_{j}\right),\bar{p}\left(y_{2}|x_{j}\right),\ldots ,\bar{p}\left(y_{\left|Y\right|}|x_{j}\right)\right], \end{array}$$
*where $\bar{p}\left(y|x_{j}\right)$ is given by*
(10)$$\begin{array}{c} \bar{p}\left(y|x_{j}\right)=\frac{1}{n}\sum_{k=1}^{n}p_{{}_{k}}\left(y|x_{j}\right). \end{array}$$

*Then, for a given r≥1, the error probability of the proposed classifier is upper bounded by*
(11)$$\begin{array}{c} P_{e}\leq 2|Y|e^{-2n\epsilon^{2}}+2\left|Y\right|e^{-2nt^{1/3}\epsilon^{2}}, \end{array}$$
*where ϵ is given by*
(12)$$\begin{array}{c} \epsilon =\min_{\begin{array}{c} i,j\\ i\neq j \end{array}}\frac{{∥P_{{\hat{y}}_{i}^{nt}}-{\bar{P}}_{j}∥}_{r}}{\left(2+t^{-1/3}\right){\left|Y\right|}^{1/r}}. \end{array}$$


**Proof of Theorem 1.** Without loss of generality we assume that $x_{1}$ is the input to $p_{{}_{Y^{n}|X}}\left(y^{n}|x\right)$ and $y^{n}$ is observed.Let $A_{k}^{\epsilon }$, for 1≤k≤|Y|, be a set defined as
(13)$$\begin{array}{c} A_{k}^{\epsilon }=\left\{y^{n}:|\frac{I\left[y^{n}=y_{k}\right]}{n}-\bar{p}\left(y_{k}|x_{1}\right)|\leq \epsilon \right\}. \end{array}$$Furthermore, let $B_{k}^{\epsilon }$, for 1≤k≤|Y|, be a set defined as
(14)$$\begin{array}{c} B_{k}^{\epsilon }=\left\{{\hat{y}}^{nt}:|\frac{I\left[{\hat{y}}^{nT}=y_{k}\right]}{nt}-\bar{p}\left(y_{k}|x_{1}\right)|\leq \frac{\epsilon }{\sqrt[{3}]{t}}\right\}. \end{array}$$Let $A^{\epsilon }=\overset{\left|Y\right|}{\underset{k=1}{⋂}}A_{k}^{\epsilon }$ and $B^{\epsilon }=\overset{\left|Y\right|}{\underset{k=1}{⋂}}B_{k}^{\epsilon }$. Now, for any $y^{n}\in A^{\epsilon }$, we have
(15)$$\begin{array}{c} {\left(\sum_{k=1}^{\left|Y\right|}|\frac{I[y^{n}=y_{k}]}{n}-\bar{p}\left(y_{k}|x_{1}\right){|}^{r}\right)}^{1/r}\overset{\left(a\right)}{\leq }{\left(\sum_{k=1}^{\left|Y\right|}\epsilon^{r}\right)}^{1/r}, \end{array}$$
where (a) follows from (Equation 13). Moreover, for ${\hat{y}}_{1}^{nt}\in B^{\epsilon }$, we have
(16)$$\begin{array}{c} {\left(\sum_{k=1}^{\left|Y\right|}|\frac{I[{\hat{y}}_{1}^{nt}=y_{k}]}{nt}-\bar{p}\left(y_{k}|x_{1}\right){|}^{r}\right)}^{1/r}\overset{\left(a\right)}{\leq }{\left(\sum_{k=1}^{\left|Y\right|}{\left(\frac{\epsilon }{\sqrt[{3}]{t}}\right)}^{r}\right)}^{1/r}, \end{array}$$
where (a) follows from (Equation 14). Next, we have the following upper bound
(17)$$\begin{array}{cc} \; & {\left(\sum_{k=1}^{\left|Y\right|}|\frac{I[y^{n}=y_{k}]}{n}-\frac{I[{\hat{y}}_{1}^{nt}=y_{k}]}{nt}{|}^{r}\right)}^{1/r}\\ & ={\left(\sum_{k=1}^{\left|Y\right|}|\frac{I[y^{n}=y_{k}]}{n}-\bar{p}\left(y_{k}|x_{1}\right)-\left(\frac{I[{\hat{y}}_{1}^{nt}=y_{k}]}{nt}-\bar{p}\left(y_{k}|x_{1}\right)\right){|}^{r}\right)}^{1/r}\\ & \overset{\left(a\right)}{\leq }{\left(\sum_{k=1}^{\left|Y\right|}|\frac{I[y^{n}=y_{k}]}{n}-\bar{p}\left(y_{k}|x_{1}\right){|}^{r}\right)}^{1/r}+{\left(\sum_{k=1}^{\left|Y\right|}|\frac{I[{\hat{y}}_{1}^{nt}=y_{k}]}{nt}-\bar{p}\left(y_{k}|x_{1}\right){|}^{r}\right)}^{1/r}, \end{array}$$
where (a) follows from the Minkowski inequality. Combining (Equation 15)–(Equation 17), we obtain
(18)$$\begin{array}{c} {\left(\sum_{k=1}^{\left|Y\right|}|\frac{I[y^{n}=y_{k}]}{n}-\frac{I[{\hat{y}}_{1}^{nt}=y_{k}]}{nt}{|}^{r}\right)}^{1/r}\leq {\left|Y\right|}^{1/r}\epsilon +{\left|Y\right|}^{1/r}\frac{\epsilon }{\sqrt[{3}]{t}}. \end{array}$$Hence, the Minkowski distance between the empirical probability distribution of the observed vector $y^{n}$ and the empirical probability distribution of the concatenated training vector for label $x_{1}$ is upper bounded by the right hand side of (Equation 18). We now derive a lower bound for ${\hat{y}}_{i}^{nt}$, where i≠1. For any $x_{i}$, such that i≠1, we have
(19)$$\begin{array}{cc} \; & {\left(\sum_{k=1}^{\left|Y\right|}|\frac{I[y^{n}=y_{k}]}{n}-\frac{I[{\hat{y}}_{i}^{nt}=y_{k}]}{nt}{|}^{r}\right)}^{1/r}+{\left(\sum_{k=1}^{\left|Y\right|}\epsilon^{r}\right)}^{1/r}\\ \; & \overset{\left(a\right)}{\geq }{\left(\sum_{k=1}^{\left|Y\right|}|\frac{I[y^{n}=y_{k}]}{n}-\frac{I[{\hat{y}}_{i}^{nt}=y_{k}]}{nt}{|}^{r}\right)}^{1/r}+{\left(\sum_{k=1}^{\left|Y\right|}|\frac{I[y^{n}=y_{k}]}{n}-\bar{p}\left(y_{k}|x_{1}\right){|}^{r}\right)}^{1/r}\\ \; & \overset{\left(b\right)}{\geq }{\left(\sum_{k=1}^{\left|Y\right|}|\frac{I[{\hat{y}}_{i}^{nt}=y_{k}]}{nt}-\bar{p}\left(y_{k}|x_{1}\right){|}^{r}\right)}^{1/r}, \end{array}$$
where (a) follows from (Equation 15) and (b) is again due to the Minkowski inequality. The expression in (Equation 19), can be written equivalently as
(20)$$\begin{array}{cc} \; & {\left(\sum_{k=1}^{\left|Y\right|}|\frac{I[y^{n}=y_{k}]}{n}-\frac{I[{\hat{y}}_{i}^{nt}=y_{k}]}{nt}{|}^{r}\right)}^{1/r}\\ & \geq {\left(\sum_{k=1}^{\left|Y\right|}|\frac{I[{\hat{y}}_{i}^{nt}=y_{k}]}{nt}-\bar{p}\left(y_{k}|x_{1}\right){|}^{r}\right)}^{1/r}-{\left|Y\right|}^{1/r}\epsilon , \end{array}$$
where i≠1. Now, using the definitions of $P_{{\hat{y}}_{i}^{nt}}$ and ${\bar{P}}_{1}$ given by (Equation 6) and (Equation 9), respectively, into (Equation 20) we can replace the expression in the right-hand side of (Equation 20) by ${∥P_{{\hat{y}}_{i}^{nt}}-{\bar{P}}_{1}∥}_{r}$, and thereby for any i≠1 we have
(21)$$\begin{array}{cc} {\left(\sum_{k=1}^{\left|Y\right|}|\frac{I[y^{n}=y_{k}]}{n}-\frac{I[{\hat{y}}_{i}^{nt}=y_{k}]}{nt}{|}^{r}\right)}^{1/r}\geq & {∥P_{{\hat{y}}_{i}^{nt}}-{\bar{P}}_{1}∥}_{r}-{\left|Y\right|}^{1/r}\epsilon . \end{array}$$The expression in (Equation 21) represents a lower bound on the Minkowski *r* distance between the empirical probability distribution of the observed vector $y^{n}$ and the empirical probability distribution of the concatenated training vector for any label $x_{i}$, where i≠1.Using the bounds in (Equation 18) and (Equation 21), we now relate the left-hand sides of (Equation 18) and (Equation 21). As long as the following inequality holds for each i≠1,
(22)$$\begin{array}{c} {\left|Y\right|}^{1/r}\epsilon \left(1+\frac{1}{\sqrt[{3}]{t}}\right)<∥P_{{\hat{y}}_{i}^{nt}}-{\bar{P}}_{1}{∥}_{r}-{\left|Y\right|}^{1/r}\epsilon , \end{array}$$
which is equivalent to the following for i≠1
(23)$$\begin{array}{c} \epsilon <\frac{{∥P_{{\hat{y}}_{i}^{nt}}-{\bar{P}}_{1}∥}_{r}}{\left(2+t^{-1/3}\right){\left|Y\right|}^{1/r}}, \end{array}$$
(24)$$\begin{array}{cc} {\left(\sum_{k=1}^{\left|Y\right|}|\frac{I[y^{n}=y_{k}]}{n}-\frac{I[{\hat{y}}_{1}^{nt}=y_{k}]}{nt}{|}^{r}\right)}^{1/r} & \overset{\left(a\right)}{\leq }{\left|Y\right|}^{1/r}\epsilon \left(1+\frac{1}{\sqrt[{3}]{t}}\right)\\ \; & \overset{\left(b\right)}{<}∥P_{{\hat{y}}_{i}^{nt}}-{\bar{P}}_{1}{∥}_{r}-|Y{|}^{1/r}\epsilon \\ & \overset{\left(c\right)}{\leq }{\left(\sum_{k=1}^{\left|Y\right|}|\frac{I[y^{n}=y_{k}]}{n}-\frac{I[{\hat{y}}_{i}^{nt}=y_{k}]}{nt}{|}^{r}\right)}^{1/r}, \end{array}$$
where (a), (b), and (c) follow from (Equation 18), (Equation 22), and (Equation 21), respectively. Thereby, from (Equation 24), we have the following for i≠1
(25)$$\begin{array}{c} {\left(\sum_{k=1}^{\left|Y\right|}|\frac{I[y^{n}=y_{k}]}{n}-\frac{I[{\hat{y}}_{1}^{nT}=y_{k}]}{nT}{|}^{r}\right)}^{1/r}<{\left(\sum_{k=1}^{\left|Y\right|}|\frac{I[y^{n}=y_{k}]}{n}-\frac{I[{\hat{y}}_{i}^{nT}=y_{k}]}{nT}{|}^{r}\right)}^{1/r}. \end{array}$$Note that the right- and left-hand sides of (Equation 25) can be replaced by the Minkowski distance of the vectors
(26)$$\begin{array}{c} v_{1}=\left[\frac{I\left[y^{n}=y_{1}\right]}{n}-\frac{I\left[{\hat{y}}_{1}^{nt}=y_{1}\right]}{nt},\ldots ,\frac{I\left[y^{n}=y_{\left|Y\right|}\right]}{n}-\frac{I\left[{\hat{y}}_{1}^{nt}=y_{\left|Y\right|}\right]}{nt}\right], \end{array}$$
and
(27)$$\begin{array}{c} v_{2}=\left[\frac{I\left[y^{n}=y_{1}\right]}{n}-\frac{I\left[{\hat{y}}_{i}^{nt}=y_{1}\right]}{nt},\ldots ,\frac{I\left[y^{n}=y_{\left|Y\right|}\right]}{n}-\frac{I\left[{\hat{y}}_{i}^{nt}=y_{\left|Y\right|}\right]}{nt}\right], \end{array}$$
respectively. Now, (Equation 26) and (Equation 27) can be replaced by $P_{y^{n}}-P_{{\hat{y}}_{1}^{nt}}$ and $P_{y^{n}}-P_{{\hat{y}}_{i}^{nt}}$, respectively, by the definitions of $P_{y^{n}}$ and $P_{{\hat{y}}_{i}^{nt}}$ given by (Equation 7) and (Equation 6), respectively. Therefore, (Equation 25) can be written equivalently as
(28)$$\begin{array}{c} {∥P_{y^{n}}-P_{{\hat{y}}_{1}^{nt}}∥}_{r}<{∥P_{y^{n}}-P_{{\hat{y}}_{i}^{nt}}∥}_{r}. \end{array}$$Now, let us highlight what we have obtained. We obtained that there is an ϵ for which if (Equation 23) holds for i≠1, and for that ϵ there are sets $A^{\epsilon }$ and $B^{\epsilon }$ for which $y^{n}\in A^{\epsilon }$ and ${\hat{y}}_{1}^{nt}\in B^{\epsilon }$ then (Equation 28) holds for i≠1, and thereby our classifier will detect that $x_{1}$ is the correct label. Using this, we can upper bound the error probability as
(29)$$\begin{array}{cc} P_{e} & =1-\mathrm{Pr}\left\{{\hat{x}}_{1}=x_{1}\right\}\\ & \leq 1-\mathrm{Pr}\{\left(y^{n}\in A^{\epsilon }\right)\cap \left({\hat{y}}_{1}^{nt}\in B^{\epsilon }\right)\left|\epsilon \in S\right\}, \end{array}$$
where S is a set defined as
(30)$$\begin{array}{c} S=\left\{\epsilon :\epsilon \leq \min_{\begin{array}{c} i\\ i\neq 1 \end{array}}\frac{{∥P_{{\hat{y}}_{i}^{nt}}-{\bar{P}}_{1}∥}_{r}}{\left(2+t^{-1/3}\right){\left|Y\right|}^{1/r}}\right\}. \end{array}$$In the following, we derive the expression in (Equation 29). The right-hand side of (Equation 29) can be upper bounded as -4.6cm0cm
(31)$$\begin{array}{cc} 1-\mathrm{Pr}\{\left(y^{n}\in A^{\epsilon }\right)\cap \left({\hat{y}}_{1}^{nt}\in B^{\epsilon }\right)\left|\epsilon \in S\right\} & =\mathrm{Pr}\{\left(y^{n}\notin A^{\epsilon }\right)\cup \left({\hat{y}}_{1}^{nt}\notin B^{\epsilon }\right)\left|\epsilon \in S\right\}\\ \; & \overset{\left(a\right)}{\leq }\mathrm{Pr}\left\{y^{n}\notin A^{\epsilon }|\epsilon \in S\right\}+\mathrm{Pr}\{{\hat{y}}_{1}^{nt}\notin B^{\epsilon }\left|\epsilon \in S\right\}, \end{array}$$
where (a) follows from Boole’s inequality. Now, note that we have the following upper bound for the first expression in the right-hand side of (Equation 31)
(32)$$\begin{array}{cc} \mathrm{Pr}\left\{y^{n}\notin A^{\epsilon }|\epsilon \in S\right\} & =\mathrm{Pr}\{y^{n}\notin \overset{\left|Y\right|}{\underset{k=1}{⋂}}A_{k}^{\epsilon }\left|\epsilon \in S\right\}\\ \; & =\mathrm{Pr}\{y^{n}\in \overset{\left|Y\right|}{\underset{k=1}{⋃}}\bar{A_{k}^{\epsilon }}\left|\epsilon \in S\right\}\\ & \overset{\left(a\right)}{\leq }\sum_{k=1}^{\left|Y\right|}\mathrm{Pr}\left\{y^{n}\in \bar{A_{k}^{\epsilon }}|\epsilon \in S\right\}\\ & =\sum_{k=1}^{\left|Y\right|}\mathrm{Pr}\{|\frac{I[y^{n}=y_{k}]}{n}-\bar{p}\left(y_{k}|x_{1}\right)|>\epsilon \left|\epsilon \in S\right\}\\ & =\sum_{k=1}^{\left|Y\right|}\mathrm{Pr}\{|\sum_{j=1}^{n}\frac{Z[y_{j}=y_{k}]}{n}-\bar{p}\left(y_{k}|x_{1}\right)|>\epsilon \left|\epsilon \in S\right\}, \end{array}$$
where $\bar{A_{k}^{\epsilon }}$ is the complement of $A_{k}^{\epsilon }$ and (a) follows from Boole’s inequality. Note that $Z\left[y_{1}=y_{k}\right],Z\left[y_{2}=y_{k}\right],\ldots ,Z\left[y_{n}=y_{k}\right]$ in (Equation 32) are *n* independent Bernoulli random variables with probabilities of success $p_{{}_{1}}\left(y_{k}|x_{1}\right),p_{{}_{2}}\left(y_{k}|x_{1}\right),\ldots ,p_{{}_{n}}\left(y_{k}|x_{1}\right)$, respectively. Let $W[y_{k}]$ be a binomial random variable with parameters $\left(n,\bar{p}\left(y_{k}|x_{1}\right)\right)$. We proceed the proof by introducing the following well-known Hoefdding’s Theorem from [41]. □

**Theorem** **2**(Hoeffding [41])**.**
*Assume that $Z_{1},Z_{2},\ldots ,$ and $Z_{n}$ are n independent Bernoulli random variables with probabilities of success $p_{{}_{1}},p_{{}_{2}},\ldots ,$ and $p_{{}_{n}}$, respectively. Next, let Z be defined as $Z=Z_{1}+Z_{2}+\ldots +Z_{n}$ and, let $\bar{p}$ be defined as $\bar{p}=\left(p_{{}_{1}}+p_{{}_{2}}+\ldots +p_{{}_{n}}\right)/n$. Let W be a binomial random variable with parameters $\left(n,\bar{p}\right)$. Then, for a given a and b, where $0\leq a\leq n\bar{p}\leq b\leq n$ holds, we have*
(33)$$\begin{array}{c} \mathrm{Pr}\left\{a\leq W\leq b\right\}\leq \mathrm{Pr}\left\{a\leq Z\leq b\right\}. \end{array}$$
*In other words, the probability distribution of W is more dispersed around its mean $n\bar{p}$ than is the probability distribution of Z. Except in the trivial case when a=b=0, the bound in *(Equation 33)* holds with equality if and only if $p_{1}=\ldots =p_{n}=\bar{p}$.*

**Proof of Theorem 2.** Please refer to [41]. □

Setting $a=n(\bar{p}-\delta )$ and $b=n(\bar{p}+\delta )$ in (Equation 33), we obtain
(34)$$\begin{array}{c} \mathrm{Pr}\left\{n\left(\bar{p}-\delta \right)\leq W\leq n\left(\bar{p}+\delta \right)\right\}\leq \mathrm{Pr}\left\{n\left(\bar{p}-\delta \right)\leq Z\leq n\left(\bar{p}+\delta \right)\right\}. \end{array}$$

Using (Equation 34), we have the following upper bound
(35)$$\begin{array}{cc} \mathrm{Pr}\left\{|\frac{Z}{n}-\bar{p}|>\delta \right\} & =1-\mathrm{Pr}\left\{n\left(\bar{p}-\delta \right)\leq Z\leq n\left(\bar{p}+\delta \right)\right\}\\ & \overset{\left(a\right)}{\leq }1-\mathrm{Pr}\left\{n\left(\bar{p}-\delta \right)\leq W\leq n\left(\bar{p}+\delta \right)\right\}\\ & =\mathrm{Pr}\left\{|\frac{W}{n}-\bar{p}|>\delta \right\}, \end{array}$$
where (a) follows from (Equation 34).

We now turn to the proof of Theorem 1. According to Theorem 2, the probability distribution of $W[y_{k}]$ is more dispersed around its mean $n\bar{p}\left(y_{k}|x_{1}\right)$ than is the probability distribution of $\sum_{1\leq j\leq n}Z\left[y_{j}=y_{k}\right]$. Therefore, we can upper bound the probability in the last line of (Equation 32) as
(36)$$\begin{array}{c} \mathrm{Pr}\{|\sum_{j=1}^{n}\frac{Z[y_{j}=y_{k}]}{n}-\bar{p}\left(y_{k}|x_{1}\right)|>\epsilon \left|\epsilon \in S\right\}\overset{\left(a\right)}{\leq }\mathrm{Pr}\{|\frac{W[y_{k}]}{n}-\bar{p}\left(y_{k}|x_{1}\right)|>\epsilon |\epsilon \in S\}, \end{array}$$
where ϵ∈S is defined in (Equation 30) and (a) follows from (Equation 35). Now, let us introduce another well-known Hoeffding’s Theorem from [42].

**Theorem** **3**(Hoeffding’s inequality [42])**.**
*Let $W_{1},W_{2},\ldots ,W_{n}$ be n independent random variables such that for each 1≤i≤n, we have $\mathrm{Pr}\left\{W_{i}\in \left[a_{i},b_{i}\right]\right\}=1$. Then for $S_{n}$, defined as $S_{n}=\sum_{i=1}^{n}W_{i}$, we have*
(37)$$\begin{array}{c} \mathrm{Pr}\left\{S_{n}-E\left[S_{n}\right]\geq \delta \right\}\leq \exp\left(-\frac{2\delta^{2}}{\sum_{i=1}^{n}{\left(b_{i}-a_{i}\right)}^{2}}\right), \end{array}$$
*where $E\left[S_{n}\right]$ is the expectation of $S_{n}$.*

**Proof of Theorem 3.** Please refer to [42]. □

Back to (Equation 36), by using the result of (Equation 37) for $a_{i}=0$ and $b_{i}=1$ since the binomial random variable $W[y_{k}]$ can take values 0 or 1, respectively, we have
(38)$$\begin{array}{cc} \mathrm{Pr}\{|\sum_{j=1}^{n}\frac{Z[y_{j}=y_{k}]}{n}-\bar{p}\left(y_{k}|x_{1}\right)|>\epsilon |\epsilon \in S\} & \leq 2\exp\left(-\frac{2n^{2}\epsilon^{2}}{\sum_{1\leq i\leq n}{\left(1-0\right)}^{2}}\right)\\ \; & \leq 2e^{-2n\epsilon^{2}}, \end{array}$$
where ϵ∈S is defined in (Equation 30). Inserting (Equation 38) into (Equation 32), we obtain the following upper bound
(39)$$\begin{array}{c} \mathrm{Pr}\left\{y^{n}\notin A^{\epsilon }|\epsilon \in S\right\}\leq 2\left|Y\right|e^{-2n\epsilon^{2}}. \end{array}$$

Similarly, we have the following result for the second expression in the right-hand side of (Equation 31)
(40)$$\begin{array}{cc} \mathrm{Pr}\{{\hat{y}}_{1}^{nt}\notin B^{\epsilon }|\epsilon \in S\} & =\mathrm{Pr}\{{\hat{y}}_{1}^{nt}\notin \overset{\left|Y\right|}{\underset{k=1}{⋂}}B_{k}^{\epsilon }|\epsilon \in S\}\\ \; & =\mathrm{Pr}\{{\hat{y}}_{1}^{nt}\in \overset{\left|Y\right|}{\underset{k=1}{⋃}}\bar{B_{k}^{\epsilon }}|\epsilon \in S\}\\ & \overset{\left(a\right)}{\leq }\sum_{k=1}^{\left|Y\right|}\mathrm{Pr}\left\{{\hat{y}}_{1}^{nt}\in \bar{B_{k}^{\epsilon }}|\epsilon \in S\right\}\\ & =\sum_{k=1}^{\left|Y\right|}\mathrm{Pr}\{|\frac{I[{\hat{y}}_{1}^{nt}=y_{k}]}{nt}-\bar{p}\left(y_{k}|x_{1}\right)|>\frac{\epsilon }{\sqrt[{3}]{t}}|\epsilon \in S\}\\ & =\sum_{k=1}^{\left|Y\right|}\mathrm{Pr}\{|\sum_{j=1}^{nt}\frac{Z[y_{j}=y_{k}]}{nt}-\bar{p}\left(y_{k}|x_{1}\right)|>\frac{\epsilon }{\sqrt[{3}]{t}}|\epsilon \in S\}, \end{array}$$
where again (a) follows from Boole’s inequality. Note that due to (Equation 5), for any integer number *l* such that 0≤l≤t−1 the random variables $Z\left[y_{nl+1}=y_{k}\right],Z\left[y_{nl+2}=y_{k}\right],\ldots ,$ and $Z[y_{nl+n}=y_{k}]$ in (Equation 40) are *n* independent Bernoulli random variables with the probabilities of success $p_{{}_{1}}\left(y_{k}|x_{1}\right),p_{{}_{2}}\left(y_{k}|x_{1}\right),\ldots ,$ and $p_{{}_{n}}\left(y_{k}|x_{1}\right)$, respectively $(y_{nl+1},y_{nl+2},\ldots ,y_{nl+n}$ are elements of ${\hat{y}}_{1_{l+1}}^{n})$. In addition, note that
(41)$$\begin{array}{cc} \bar{p}\left(y_{k}|x_{1}\right) & =\frac{1}{n}\sum_{j=1}^{n}p_{{}_{j}}\left(y_{k}|x_{1}\right)\\ & =\frac{1}{nt}\left(\sum_{l=0}^{t-1}\sum_{j=1}^{n}p_{{}_{j}}\left(y_{k}|x_{1}\right)\right). \end{array}$$

Notice that for each 0≤l≤t−1, $p_{{}_{1}}\left(y_{k}|x_{1}\right)+p_{{}_{2}}\left(y_{k}|x_{1}\right)+\ldots +p_{{}_{n}}\left(y_{k}|x_{1}\right)$ is the summation of the probabilities of success of the random variables $Z\left[y_{nl+1}=y_{k}\right],Z\left[y_{nl+2}=y_{k}\right],\ldots ,$ and $Z[y_{nl+n}=y_{k}]$. Thereby, the last expression on the right-hand side of (Equation 41) is the average probability of success of random variables $Z[y_{j}=y_{k}]$ for 1≤j≤nt. Now, let $W[y_{k}]$ be a binomial random variable with parameters $\left(nt,\bar{p}\left(y_{k}|x_{1}\right)\right)$. Once again, according to Theorem 2, the probability distribution of $W[y_{k}]$ is more dispersed around its mean $nt\bar{p}\left(y_{k}|x_{1}\right))$ than is the probability distribution of $\sum_{1\leq j\leq nt}Z\left[y_{j}=y_{k}\right]$. Therefore, the probability in the last line of (Equation 40) can be upper bounded as
(42)$$\begin{array}{cc} \mathrm{Pr}\{|\sum_{j=1}^{nt}\frac{Z[y_{j}=y_{k}]}{nt}-\bar{p}\left(y_{k}|x_{1}\right)|>\frac{\epsilon }{\sqrt[{3}]{t}}|\epsilon \in S\} & \overset{\left(a\right)}{\leq }\mathrm{Pr}\{|\frac{W[y_{k}]}{nt}-\bar{p}\left(y_{k}|x_{1}\right)|>\frac{\epsilon }{\sqrt[{3}]{t}}|\epsilon \in S\}\\ \; & \overset{\left(b\right)}{\leq }2\exp\left(-\frac{2{\left(nt\right)}^{2}{\left(t^{-1/3}\epsilon \right)}^{2}}{\sum_{1\leq i\leq nt}{\left(1-0\right)}^{2}}\right)\\ \; & \leq 2e^{-2nt\left(t^{-2/3}\epsilon^{2}\right)}\\ \; & =2e^{-2nt^{1/3}\epsilon^{2}}, \end{array}$$
where ϵ∈S, defined in (Equation 30), (a) follows from (Equation 35) (in which *n* is replaced by nt), and (b) is the result of (Equation 37) for $a_{i}=0$ and $b_{i}=1$ since the binomial random variable $W[y_{k}]$ can take values 0 or 1, respectively. Inserting (Equation 42) into (Equation 40), we have the following upper bound
(43)$$\begin{array}{c} \mathrm{Pr}\{{\hat{y}}_{1}^{nt}\notin B^{\epsilon }|\epsilon \in S\}\leq 2\left|Y\right|e^{-2nt^{1/3}\epsilon^{2}}. \end{array}$$

Inserting (Equation 39) and (Equation 43) into (Equation 31), and then inserting (Equation 31) into (Equation 29), we obtain the following upper bound for the error probability
(44)$$\begin{array}{c} P_{e}\leq 2|Y|e^{-2n\epsilon^{2}}+2\left|Y\right|e^{-2nt^{1/3}\epsilon^{2}}, \end{array}$$
where
(45)$$\begin{array}{c} \epsilon =\min_{\begin{array}{c} i,j\\ i\neq j \end{array}}\frac{{∥P_{{\hat{y}}_{i}^{nt}}-{\bar{P}}_{j}∥}_{r}}{\left(2+t^{-1/3}\right){\left|Y\right|}^{1/r}}, \end{array}$$
which is the optimal value of ϵ that exhibits the tightest upper bound for the error probability $P_{e}$ given by (Equation 44). This completes the proof of Theorem 1.

The following corollary provides a simplified upper bound on the error probability when t→∞.

**Corollary** **1.**
*When the number of training vectors per label reaches infinity, i.e., when t→∞, which is equivalently to the case when the probability distribution $p(y^{n}|x)$ is known at the classifier, the error probability of the proposed classifier is upper bounded as*
(46)$$\begin{array}{c} P_{e}\leq 2\left|Y\right|e^{-2n\epsilon^{2}}, \end{array}$$
*where ϵ is given by*
(47)$$\begin{array}{c} \epsilon =\min_{\begin{array}{c} i,j\\ i\neq j \end{array}}\frac{{∥{\bar{P}}_{i}-{\bar{P}}_{j}∥}_{r}}{{2|Y|}^{1/r}}. \end{array}$$


**Proof.** The proof is straightforward. □

As can be seen from (Equation 8) and (Equation 11), the performance of the proposed classifier depends on *r*. We cannot derive the optimal value of *r* that minimizes the error probability since we do not have the exact expression of the error probability, we only have its upper bound. On the other hand, in practice, the optimal *r* with respect to the upper bound on the error probability also cannot be derived since the upper bound depends on ${\bar{P}}_{j}$, which would be unknown in practice due to $p_{Y^{n}|X}\left(y^{n}|x\right)$ being unknown. As a result, for our numerical examples, we consider the Euclidean distance (r=2), which is one of the most widely used distance metrics in practice.

The following corollary establishes the asymptotic optimality of the proposed classifier with respect to *n*.

**Corollary** **2.**
*The proposed classifier has an error probability that satisfies $P_{e}\to 0$ as n→∞ if $\left|Y|\leq O(n^{m}\right)$, m is fixed, and r>2m. Here, $n^{m}$ indicates the dimension of our space, i.e., maximum number of alphabets each element in the feature vector $y^{n}$ can take. Thereby, the proposed classifier is asymptotically optimal.*


**Proof.** For the proof, please see Appendix A. □

## 4. Simulation Results

In this section, we provide simulation results of the performance of the proposed classifier for r=2 and compare it to benchmark schemes. The benchmark schemes that we adopt for comparison are the naive Bayes classifier and the KNN algorithm. We cannot adopt a classifier based on a neural network since neural networks require a very large training set, which we assume is not available. For the naive Bayes classifier, the probability distribution $p_{{}_{Y^{n}|X}}\left(y^{n}|x\right)$ is estimated from the training vectors as follows. Let again ${\hat{y}}_{i}^{nt}$ be a vector obtained by concatenating all training feature vectors for the input label $x_{i}$ as in (Equation 5). Then, the estimated probability distribution of $p(y_{j}=y|x_{i})$, denoted by $\hat{p}\left(y_{j}=y|x_{i}\right)$, is found as
(48)$$\begin{array}{c} \hat{p}\left(y_{j}=y|x_{i}\right)=\frac{I\left[{\hat{y}}_{i}^{nt}=y\right]}{nt}, \end{array}$$
and the naive Bayes classifier decides according to
(49)$$\begin{array}{c} \hat{x}=\arg\max_{x_{i}}\prod_{k=1}^{n}\hat{p}\left(y_{k}|x_{i}\right). \end{array}$$

The main problem of the naive Bayes classifier occurs when an alphabet $y_{j}\in Y$ is not present in the training feature vectors. In that case, $\hat{p}\left(y_{j}|x_{i}\right)$ in (Equation 48) is $\hat{p}\left(y_{j}|x_{i}\right)=0$, $\forall x_{i}\in X$ and, as a result, the right hand side of (Equation 49) is zero since at least one of the elements in the product in (Equation 49) is zero. In this case, the naive Bayes classifier fails to provide an accurate classification of the labels. In what follows, we see that this issue of the naive Bayes classifier appears frequently when we have a small number of training feature vectors. On the other hand, the KNN classifier works as follows. For the observed feature vector $y^{n}$, the KNN classifier looks for the *k* nearest feature vectors to $y^{n}$, among all training feature vectors ${\hat{y}}_{r_{s}}^{n}$, for all 1≤r≤|X| and 1≤s≤T. Then by considering a set of *K* input–output pairs $\left(x_{k},{\hat{y}}_{k_{l}}^{n}\right)$, for k∈{1,2,…,|X|} and l∈{1,2,…,|T|}, the KNN classifier decides a label which is the most frequent among $x_{k}$-s. The optimum value of *k* for t=1 is k=1.

In the following, we provide numerical examples where we illustrate the performance of the proposed classifier when $p_{{}_{Y^{n}|X}}\left(y^{n}|x\right)$ is artificially generated.

### 4.1. The I.I.D. Case with One Training Sample per Label

In the following examples, we assume that the classifiers have access to only one training feature vector for each label, the elements of the feature vectors are generated i.i.d., and the alphabet size of the feature vector, |Y|, is fixed.

In Figure 4 and Figure 5, we compare the error probability of the proposed classifier with the naive Bayes classifier and the KNN algorithm for the case when |Y|=6 and |Y|=20, respectively. In both examples, we have two different labels, i.e., |X|=2. As a result, we have two different probability distributions $p_{{}_{Y^{n}|X_{1}}}\left(y^{n}|x_{1}\right)$ and $p_{{}_{Y^{n}|X_{2}}}\left(y^{n}|x_{2}\right)$. The probability distributions $p_{{}_{Y^{n}|X_{1}}}\left(y^{n}|x_{1}\right)$ and $p_{{}_{Y^{n}|X_{2}}}\left(y^{n}|x_{2}\right)$ are randomly generated as follows. We first generate two random vectors of length 6 and length 20 for Figure 4 and Figure 5, respectively, where the elements of these vectors are drawn independently from a uniform probability distribution. Then we normalize these vectors such that the sum of their elements is equal to one. These two normalized randomly generated vectors then represent the two probability distributions $p_{{}_{Y_{i}|X_{1}}}\left(y_{i}|x_{1}\right)=p_{{}_{Y|X_{1}}}\left(y|x_{1}\right)$ and $p_{{}_{Y_{i}|X_{2}}}\left(y_{i}|x_{2}\right)=p_{{}_{Y|X_{2}}}\left(y|x_{2}\right)$, ∀i. Then, $p_{{}_{Y^{n}|X_{k}}}\left(y^{n}|x_{k}\right)$ is obtained as $p_{{}_{Y^{n}|X_{k}}}\left(y^{n}|x_{k}\right)=\prod_{i=1}^{n}p_{{}_{Y_{i}|X_{k}}}\left(y_{i}|x_{k}\right)$, for k=1,2. The simulation is carried out as follows. For each *n*, we generate one training vector for each label, using the aforementioned probability distributions. Then, as test samples, we generate 1000 feature vectors for each label and pass these feature vectors through our proposed classifier, the naive Bayes classifier, and the KNN algorithm, and compute the errors. The length of the feature vector *n* is varied from n=1 to n=100. We repeat the simulation 5000 times and then plot the error probability. Figure 4 and Figure 5 show that the proposed classifier outperforms both the naive Bayes classification and KNN. The main reason for this performance gain is because when only one training vector per label is available, the proposed classifier is more resilient to errors than the naive Bayes classifier, whereas the KNN algorithm has very poor performance because of the “curse of dimensionality”. Specifically, the naive Bayes classifier cannot perform an accurate classification for small *n* compared to |Y| since the chance that an alphabet will not be present in one of the training feature vectors is close to 1. On the other hand, the KNN algorithm cannot perform an accurate classification for large *n* since the dimension of the input feature vector becomes much larger than the training data and the “curse of dimensionality” occurs.

In Figure 6, we compare the performance of the proposed classifier for different values of *r* when |Y|=6 with the derived upper bounds. As can be seen, for this example, the derived theoretical upper bounds have similar slope as the exact error probabilities. Moreover, we can see that for this example, the optimal *r* is r=1. However, this is not always the case and it depends on $p_{{}_{Y^{n}|X_{k}}}\left(y^{n}|x_{k}\right)$, |Y|, and |X|.

### 4.2. The Overlapping I.Non-I.D. Case with One Training Sample per Label

In this example, we consider the i.non-i.d. case where the probability distributions $p_{{}_{i}}\left(y_{i}|x_{k}\right)$ are overlapping for all *i*, as shown in Figure 7. The small orthogonal lines on the x-axis in Figure 7 represent alphabets, i.e., the elements in Y, and the probability of occurrence of an alphabet $y_{i}$ is equal to the intersection between the corresponding orthogonal line to the represented probability distribution $p_{{}_{i}}\left(y_{i}|x_{k}\right)$ for k=1,2. By “overlapping”, we mean the following. Let $Y_{v}$ and $Y_{u}$ denote the set of outputs generated by $p_{{}_{v}}\left(y_{v}|x_{k}\right)$ and $p_{{}_{u}}\left(y_{u}|x_{k}\right)$, respectively. If for any *v* and *u*, $Y_{v}\cap Y_{u}\neq \emptyset$ holds, we say that the output alphabets are overlapping.

To demonstrate the performance of our proposed classifier in the overlapping case, we assume that we have two different labels, $X=\{x_{1},x_{2}\}$, where the corresponding conditional probability distributions $p_{{}_{i}}\left(y_{i}|x_{1}\right)$ and $p_{{}_{i}}\left(y_{i}|x_{2}\right)$ are obtained as follows. For a given *n*, let $Y=\left\{-n,-n+1,\ldots ,0,\ldots ,n-1,n\right\}$ be the set of all alphabets. Note that the size of Y grows with *n*. Moreover, let $u_{i}$ and $v_{i}$(1≤i≤n) be vectors of length 2n+1, given by
(50)$$\begin{array}{cc} u_{i} & =\left[0,\ldots ,0,\frac{1}{i(i+1)},\frac{2}{i(i+1)},\ldots ,\frac{i}{i(i+1)},\frac{i+1}{i(i+1)},\frac{i}{i(i+1)},\ldots ,\frac{1}{i(i+1)},0,\ldots ,0\right], \end{array}$$
(51)$$\begin{array}{cc} v_{i} & =\left[0,\ldots ,0,\frac{1}{i(i+1)},\frac{1}{i(i+1)},\ldots ,\frac{1}{i(i+1)},\frac{1}{i(i+1)},0,\ldots ,0\right]. \end{array}$$

The number of zeros in each side of the vectors $u_{i}$ and $v_{i}$ is (n−i). To generate a feature vector from label $x_{1}\left(x_{2}\right)$, we generate the vector $y^{n}=\left(y_{1},y_{2},\ldots ,y_{n}\right)$, where $y_{k}$ takes values from the set Y, with a probability distribution $p_{{}_{i}}\left(y_{i}|x_{1}\right)=u_{i}\left(1+2(n+y_{i})\right)$$\left(p_{{}_{i}}\left(y_{i}|x_{2}\right)=v_{i}\left(1+2(n+y_{i})\right)\right)$.

The simulation is carried out as follows. For each *n*, we generate one training feature vector for each label. Then, we generate 1000 feature vectors for each label and pass them through our proposed classifier, the naive Bayes classifier, and the KNN algorithm and calculate the error probability. We change the length of the feature vector from n=1 to n=100 and repeat the simulation 1000 times and then plot the error probability.

As shown in Figure 8, there is a huge difference between the performance of the two benchmark classifiers and the proposed classifier. The error probability of the naive Bayes classifier is almost 0.5 for all shown values of *n* as it is susceptible to the problem of unseen alphabets in the training vectors. The error probability of the KNN classifier is also almost 0.5 for n>20 as it is susceptible to the “curse of dimensionality”. However, the error probability of our proposed classifier continuously decays as *n* increases.

In Figure 9, we run the same experiments as in Figure 8 but with T=100, i.e., 100 training feature vectors per label. As can be seen from Figure 9, the performance of the proposed classifier is better than the naive Bayes classifier, for n>15. Since |Y|=2n+1, for small values of *n*, the naive Bayes classifier has access to many training samples and, thereby, its performance is very close to the case when the probability distribution $p_{{}_{Y^{n}|X}}\left(y^{n}|x\right)$ is known, i.e., to the maximum-likelihood classifier, and hence it has the optimal performance. As *n* increases, the number of alphabets rises, i.e., |Y| rises, and due to the aforementioned issue of the naive Bayes classifier with unseen alphabets, our proposed classifier performs much better classification than the naive Bayes classifier. Furthermore, note that the error probability of our proposed classifier decays exponentially as *n* increases which is not the case with the naive Bayes classifier. Moreover, Figure 9 also shows the theoretical upper bound on the error probability we derived in (Equation 11).

### 4.3. The Non-Overlapping I.Non-I.D. Case with One Training Sample for Each Label

In this example, we consider the i.non-i.d. case where the probability distributions $p_{{}_{j}}\left(y_{j}|x_{i}\right)$ are non-overlapping for all *j* as shown in Figure 10, where we defined “overlapping” in Section 4.2. Hence, we test the other extreme in terms of possible distribution of the elements in the feature vectors $Y^{n}$.

To demonstrate the performance of our proposed classifier in the non-overlapping case, we assume that we have two different labels $X=\{x_{1},x_{2}\}$, the corresponding conditional probability distributions $p_{{}_{i}}\left(y_{i}|x_{1}\right)$ and $p_{{}_{i}}\left(y_{i}|x_{2}\right)$ are obtained as follows. For a given *n*, let $Y=\left\{1,2,3,\ldots ,{\left(n+1\right)}^{2}-1\right\}$ be the set of all alphabets of the element in the feature vectors. Note again that the size of Y grows with *n*. in addition, let $u_{i}$ and $v_{i}$ for (1≤i≤n), be vectors of length ${\left(n+1\right)}^{2}-1$, given by
(52)$$\begin{array}{cc} u_{i} & =\left[0,\ldots ,0,\frac{1}{i(i+1)},\frac{2}{i(i+1)},\ldots ,\frac{i}{i(i+1)},\frac{i+1}{i(i+1)},\frac{i}{i(i+1)},\ldots ,\frac{1}{i(i+1)},0,\ldots ,0\right], \end{array}$$
(53)$$\begin{array}{cc} v_{i} & =\left[0,\ldots ,0,\frac{1}{i(i+1)},\frac{1}{i(i+1)},\ldots ,\frac{1}{i(i+1)},\frac{1}{i(i+1)},0,\ldots ,0\right]. \end{array}$$

The number of zeros in the left-hand sides of $u_{i}$ and $v_{i}$ is $i^{2}-1$. To generate a feature vector from the label $x_{1}\left(x_{2}\right)$, we generate the vector $y^{n}=\left(y_{1},y_{2},\ldots ,y_{n}\right)$, where $y_{k}$ take values from the set Y, with probability distribution $p_{{}_{i}}\left(y_{i}|x_{1}\right)=u_{i}\left(y_{i}\right)$$\left(p_{{}_{i}}\left(y_{i}|x_{2}\right)=v_{i}\left(y_{i}\right)\right)$.

The simulation is carried out as follows. For each *n*, we generate one training feature vector for each label. Then we generate 250 feature vectors for each label and pass it through our proposed classifier, the naive Bayes classifier and KNN and calculate the error probabilities. We change the length of the vector from 1 to 80 and repeat the simulation 250 times and then plot the error probability. As shown in Figure 11, there is a huge difference between the performance of the proposed classifier and the two benchmark classifiers. The error probability of the naive Bayes classifier is almost 0.5 for all shown values of *n* as it is susceptible to the issue with unseen alphabets in the training feature vector. The error probability of the KNN classifier is almost 0.5 for all shown values of n>30 as it becomes susceptible to the “curse of dimensionality”. However, the error probability of our proposed classifier still decays continuously as *n* increases.

Note that, in our numerical examples, we compared our algorithm with the benchmark schemes on two extreme cases of i.non-i.d. vectors, referred to as “overlapping” and “non-overlapping”. Any other i.non-i.d. vector can be represented as a combination of the “overlapping” and “non-overlapping” vectors. Since our algorithm works better than the benchmark schemes for small *t* on both these cases, it will work better than the benchmark schemes on any combination between “overlapping” and “non-overlapping” vectors, i.e., for any other i.non-i.d. vectors.

## 5. Conclusions

In this paper, we proposed a supervised classification algorithm that assigns labels to input feature vectors with independent but non-identically distributed elements, a statistical property found in practice. We proved that the proposed classifier is asymptotically optimal since the error probability moves to zero as the length of the input feature vectors grows. We showed that this asymptotic optimality is achievable even when one training feature vector per label is available. In the numerical examples, we compared the proposed classifier with the naive Bayes classifier and the KNN algorithm. Our numerical results show that the proposed classifier outperforms the benchmark classifiers when the number of training data is small and the length of the input feature vectors is sufficiency large.

## Figures and Tables

**Figure 1 entropy-23-01045-f001:**
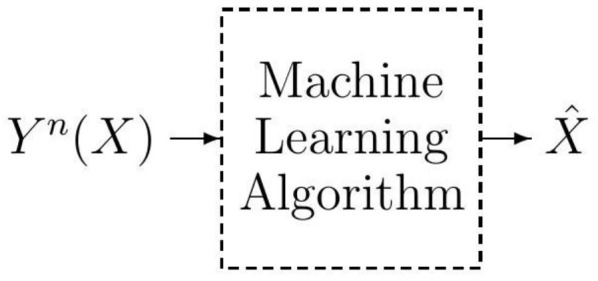
A typical structural modelling of the classification learning problem.

**Figure 2 entropy-23-01045-f002:**
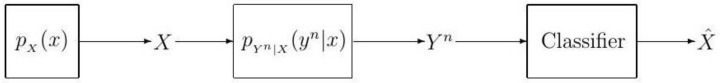
An alternative modeling of the classification learning problem.

**Figure 3 entropy-23-01045-f003:**
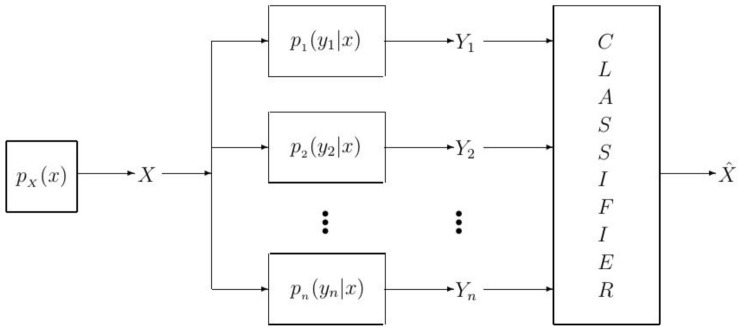
An alternative modelling of the classification learning problem when the elements of $Y^{n}\left(X\right)$ are mutually independent but non-identically distributed (i.non-i.d.).

**Figure 4 entropy-23-01045-f004:**
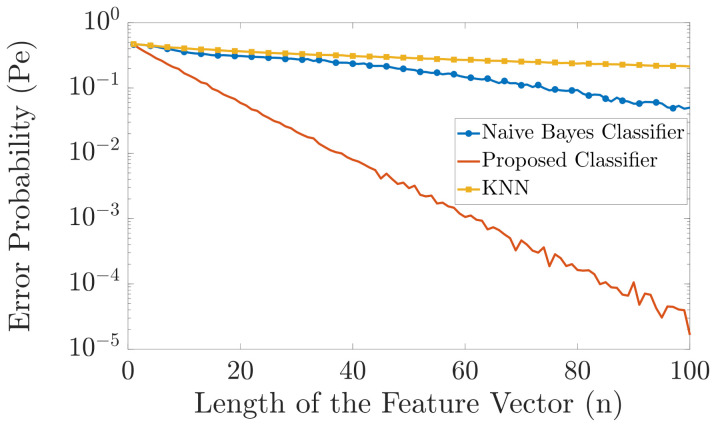
Comparison in error probability between the naive Bayes classifier, KNN, and the proposed classifier.

**Figure 5 entropy-23-01045-f005:**
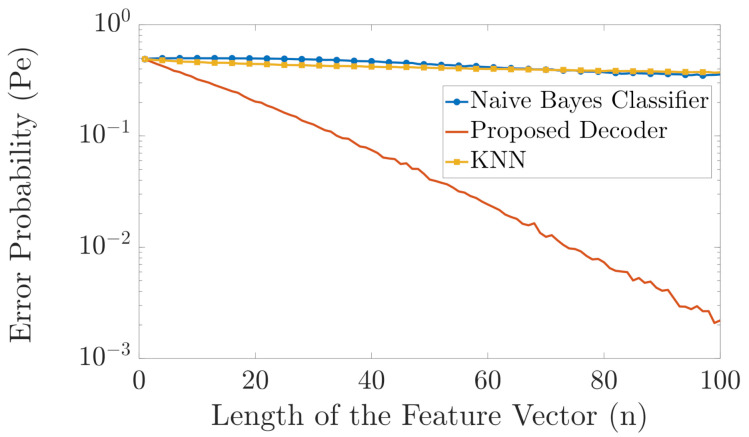
Comparison in error probability between the naive Bayes classifier, KNN, and the proposed classifier.

**Figure 6 entropy-23-01045-f006:**
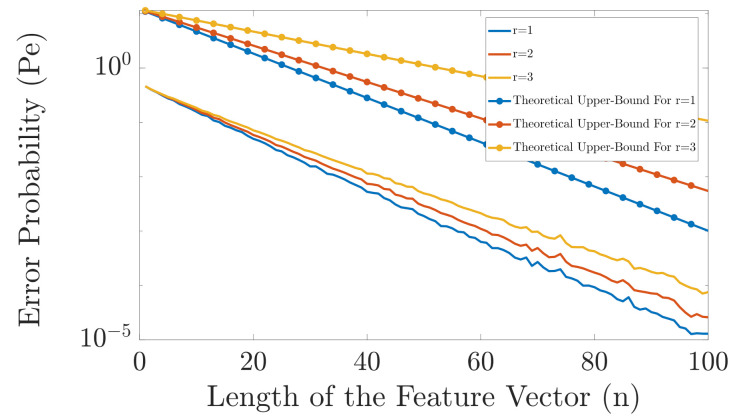
Comparison in error probability of the proposed classifier for different values of *r* when |Y|=6. The related theoretical upper bounds for each value of *r* are also given.

**Figure 7 entropy-23-01045-f007:**
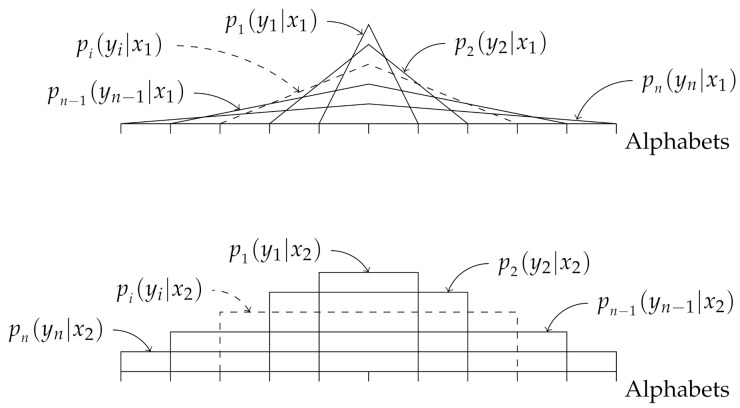
Illustration of the probability distributions $p_{{}_{i}}\left(y_{i}|x_{1}\right)$ (upper figure) and $p_{{}_{i}}\left(y_{i}|x_{2}\right)$ (lower figure), for i=1,2,…,n.

**Figure 8 entropy-23-01045-f008:**
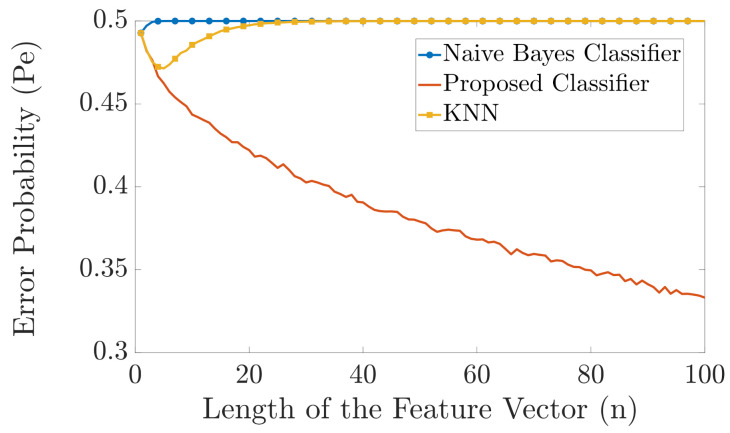
Comparison in error probability between the naive Bayes classifier, KNN, and the proposed classifier (T=1).

**Figure 9 entropy-23-01045-f009:**
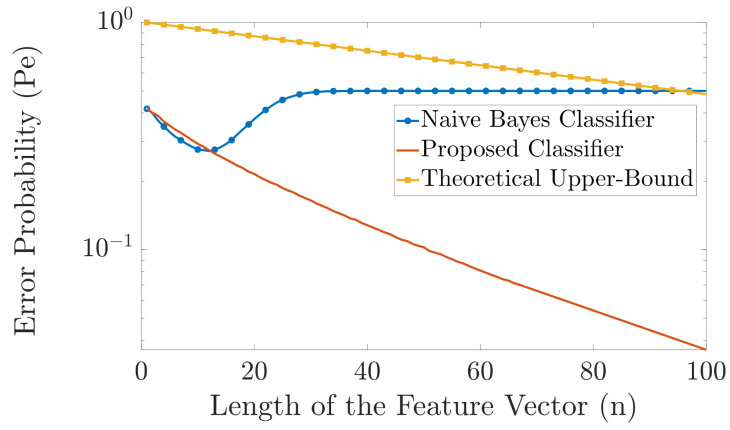
Comparison in error probability between the naive Bayes classifier and the proposed classifier (T=100).

**Figure 10 entropy-23-01045-f010:**
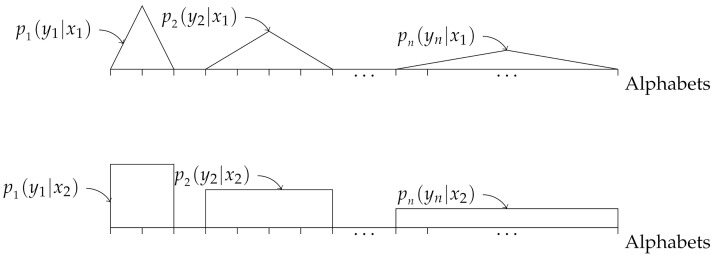
Illustration of the probability distributions $p_{{}_{i}}\left(y_{i}|x_{1}\right)$ (upper figure) and $p_{{}_{i}}\left(y_{i}|x_{2}\right)$ (lower figure), for i=1,2,…,n.

**Figure 11 entropy-23-01045-f011:**
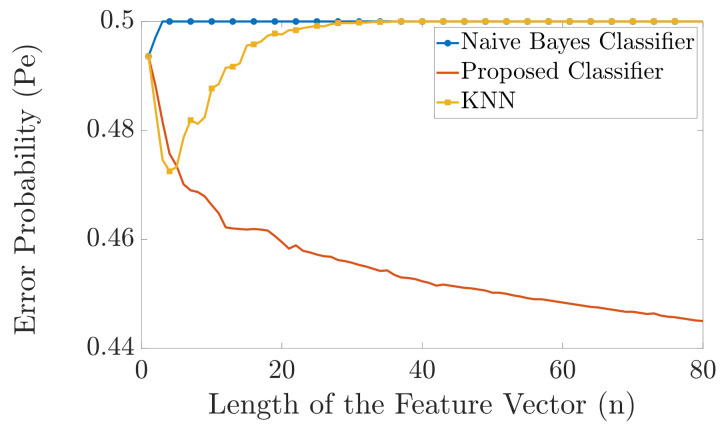
Comparison in error probability between the naive Bayes classifier and the proposed classifier (T=1).

## Data Availability

Data sharing is not applicable to this article as no new data were created or analyzed in this study.

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
