# Peer review of "On Supervised Classification of Feature Vectors with Independent and Non-Identically Distributed Elements"

_entropy, 2021, doi:10.3390/e23081045_

Round 1
Reviewer 1 Report
The work presented is interesting and well written. In my opinion, the work will be popular among readers of Entropy. I have just a few comments on the text. 1. It should be noted that only discrete random variables X are considered in the work. 2. The text before formula (24) should be corrected. 3. The beginning of the proof of Theorem 1 should be noted. 4. In my opinion, there is no need for an appendix, the proof of Corollary 2 is better to present in the main text. 5. The text of Proposition 1 should be clearer. 6. The index in formula (8) needs to be corrected. 7. It is necessary to rewrite the reference list, according to the requirements of the journal.Author Response
Please see the attachment.

Reviewer 2 Report
The paper focuses on a supervised classifier based on a nearest neighborhood procedure in terms of empiric probability distributions for i. but not i.d. elements of the feature vector. In particular, they provide an explicit upper bound on the error probability and they prove the asymptotic optimality of the proposed classifier. Finally, they show that such classifier outruns both the naive Bayes classifier and the KNN classifier (that are penalized by the fact that as the feature vector grows, it becomes much larger than the training sets).
The paper is really well written and extremely clear, in particular, in underlining the novelty not of the algorithm itself but of the upper bound on the error probability. I have only two really minor comments that I think can be applied directly in the last phase of the publication procedure:
- In Equation (25) I should use < in place of <= as given by Equation (24) and as used in (28);
- Two lines after equation (28) I suggest to add the word "if" after "for which".
As I already stated, this paper is really well written and the results are very interesting and useful. Thus, I can suggest the publication.
